# Exploring Patient Awareness and the Feasibility of Mediation in Healthcare: A Pilot Study in Bulgaria

**DOI:** 10.3390/healthcare13060629

**Published:** 2025-03-14

**Authors:** Kostadin Yordanov Dimitrov, Tsonka Miteva-Katrandzhieva

**Affiliations:** Department of Social Medicine and Public Health, Faculty of Public Health, Medical University—Plovdiv, Vasil Aprilov Blvd 15A, 4002 Plovdiv, Bulgaria; tsonka.miteva@mu-plovdiv.bg

**Keywords:** healthcare mediation, healthcare conflict resolution, physician–patient conflict, physician–patient communication, alternative dispute resolution methods

## Abstract

Background/Objectives: The healthcare system is complex and emotionally charged, which frequently leads to conflicts between patients and healthcare providers as a result of inadequate communication and unmet patient expectations. This pilot study investigates patient awareness and the feasibility of mediation as an alternative conflict resolution method in healthcare. Methods: A cross-sectional survey of 40 Bulgarian citizens was conducted to evaluate their experiences with physician–patient communication, their opinions regarding the effectiveness of the legal system, and their awareness of mediation. Results: According to the findings, more than half of the respondents reported difficulties communicating with their physicians, and only 5% believed the judicial system was effective in resolving disputes in healthcare. While many were familiar with mediation, only 2.5% had used it to address healthcare-related problems. Conclusions: This pilot study underscores the need for targeted education and trust-building initiatives to facilitate the implementation of mediation in healthcare. Improving patient–physician communication and introducing mediation could enhance healthcare quality and patients’ trust, providing a more efficient and satisfactory conflict resolution mechanism.

## 1. Introduction

The provision of healthcare is a complex and multi-layered process that incorporates a wide range of professions and services. Furthermore, the process of providing and receiving necessary healthcare services is personal and frequently provokes intense emotions in both the patient and healthcare professionals [1]. The dynamic development of the healthcare field, combined with continually increasing patient expectations, create conditions for conflicts that many consider inevitable. The specificity of this field presents health workers with numerous challenges, often leading to disagreements with patients or their relatives [2,3]. Within this context, poor communication is the most frequent reason for conflicts between physicians and patients [4,5]. Patients and their families are more likely to take legal action and enter conflicts with physicians when they do not receive enough information about their condition and treatment, feel ignored, or have their concerns unaddressed [6]. Physicians who are rude, fail to communicate effectively with patients, and show a lack of respect are at significantly higher risk of conflicts and of facing legal claims [7]. If not managed appropriately and promptly, conflicts can have a negative impact on healthcare facilities (including legal costs, higher staff turnover, and reduced productivity), healthcare professionals (burnout, strained team relationships, and poorer well-being), and most importantly, patients (medical errors, deteriorating physician–patient relationships, and poor healthcare outcomes) [8].

The rise in the number of lawsuits in healthcare over the last decade, in Bulgaria and other countries (exceeding 50% in the UK and in Baltic and Eastern European countries, and experiencing a triple-digit percentage increase of 200–500% in Germany, Italy, Iberian countries, and the Mediterranean region [9,10]) coupled with the prolonged duration of legal proceedings, the inefficiencies of the judicial system in terms of resource allocation, its overload, and its limitations in delivering fair decisions in each case [11], indicate a growing need for an alternative solution to manage conflicts in healthcare. To address these issues, countries such as the UK, China, Singapore, and select states in the US have adopted alternative dispute resolution methods, specifically mediation [12,13,14,15].

Alternative dispute resolution methods refer to various approaches for resolving conflicts in a cooperative and non-adversarial way without involving the court system [16]. Mediation is among the most significant and effective methods of alternative dispute resolution. It is a voluntary, flexible, and confidential process in which a third party (the mediator) assists the parties in the dispute to reach a consensus without court intervention [17,18]. The main role of the mediator is to help the parties clarify their issues, explore their interests, create a safe environment of trust for discussing potentially emotionally and psychologically challenging issues, and steer the conversation towards problem resolution and consensus. Unlike a judge or a judicial jury, the mediator cannot make a final decision or take sides in the dispute [19]. The advantages of mediation include cost and time efficiency, the parties having greater control over the process, and open communication between the parties—allowing patients to freely express their concerns and views [3,20]. Mediation is particularly suitable for resolving conflicts in healthcare due to the highly emotional nature of such conflicts, which are often associated with feelings of anger, suffering, and stress. The informal and secure setting of the mediation process, facilitated by a third party, enables swift and open discussion of issues, meeting the expectations of both sides in the dispute [17].

Bulgaria introduced a law on mediation in 2004, and following the adoption of the European Parliament’s Mediation Directive 2008/52/EC, all EU member states were required to incorporate the directive into their legal systems by 21 May 2011 [21,22]. However, mediation remains a relatively new and underutilized approach in the healthcare field, particularly in Bulgaria. In contrast, countries like the UK have successfully integrated mediation into their healthcare systems. After a promising pilot program, the UK’s National Health Service formally launched mediation services for claims in December 2016. Between its launch and March 2019, 606 healthcare cases were handled, with 74% resolved within 28 days [23].

The main goal of this study is to evaluate patients’ awareness of mediation and to investigate the feasibility of introducing mediation as an alternative to the court system for resolving disputes within the healthcare system. At the time of this study, no similar research had been conducted in Bulgaria. Furthermore, the available literature does not provide insights into how mediation could be effectively incorporated into the Bulgarian healthcare system. Given the need for exploring alternative dispute resolution methods to address healthcare disputes in the country, along with the existing legislation on mediation, the gap in the literature has driven the need for this study, highlighting its importance, as mediation is still not widely practiced or studied for healthcare dispute resolution in Bulgaria.

## 2. Materials and Methods

A pilot cross-sectional study was conducted using an anonymous survey. The survey was administered online using the Lime Survey platform (LimeSurvey GmbH, n.d.) between 1 July 2024 and 1 August 2024. The study population included individuals aged 18 years and older residing in Bulgaria. To ensure comprehensive representation, the inclusion criteria were not limited to individuals currently seeking medical assistance or actively engaging with healthcare institutions. This broader approach was selected because Bulgarian legislation requires annual prophylactic medical check-ups, which ensures that all respondents have had frequent interactions with the healthcare system, shaping their own opinion about the topics included in the survey. The sole exclusion criterion was individuals under 18 years of age. In this pilot study, we selected a sample of at least 40 participants to investigate Bulgarian residents’ openness to mediation as an alternative to the traditional court system for conflict resolution. Additionally, we aimed to assess whether a lack of awareness serves as a primary barrier to the broader implementation of mediation in healthcare, or if other underlying factors contribute to this challenge. This sample size was deemed appropriate for obtaining preliminary insights and facilitating the design of a subsequent full-scale study.

For the study, a structured questionnaire in Bulgarian language was developed. The questionnaire was created by the first author, a certified mediator and medical doctor, following a comprehensive literature review. To ensure the tool’s alignment with the study’s objectives and its ability to accurately measure the intended constructs, it was validated by the second author, a certified mediator and dental doctor. The validation process focused on ensuring the questionnaire’s content relevance, clarity, and reliability in capturing the key aspects of the research. The final version of the questionnaire, consisting of 44 questions, was organized into four sections: a socio-demographic profile, which gathered participants’ demographic details; conflicts in healthcare, examining physician–patient conflicts, their underlying causes, and their impact on the quality and delivery of healthcare services (to assess the effect of physician–patient communication on perceived healthcare quality, a five-point Likert scale was included in this section, where 1 represents “no effect” and 5 represents a “strong effect”); legal system and healthcare, exploring patients’ perceptions of the legal system’s effectiveness in resolving healthcare-related disputes; and mediation in healthcare, evaluating patient awareness of mediation as a conflict resolution method, its feasibility, potential adoption barriers, preferences for mediation organization, and the likelihood of its acceptance and use.

Forty participants completed the questionnaire. The data were analyzed using IBM SPSS Statistics v. 19, employing the following statistical methods:Descriptive Statistics: Used to summarize and describe the main features of the data collected. This included calculating frequency counts and percentages for categorical variables, as well as measures of central tendency (mean) and dispersion (standard deviation) for continuous variables. Additionally, confidence intervals (95% CI) were calculated for the mean to provide an estimate of the range within which the true population mean is likely to fall.Shapiro–Wilk test: This test was used to assess whether the age variable was normally distributed. The test was performed at a significance level of *p* = 0.05.Pearson’s Chi-Square test: This test was applied to evaluate associations between gender and age with the following variables: the perceived impact of physician communication on healthcare quality, the occurrence of conflicts between physician and patient, the perceived effectiveness of the court system in resolving healthcare conflicts, the evaluation of mediation’s potential to resolve healthcare disputes, and willingness to use mediation services provided by healthcare institutions. The significance level chosen for the analysis was set at *p* = 0.05.

## 3. Results

### 3.1. Sociodemographic Characteristics of Respondents

Of the 40 respondents, women were more prevalent. The Shapiro–Wilk test indicated that the age variable was normally distributed (*p* = 0.230, df = 39). Participants’ ages ranged from 21 to 80 years, with a mean age of 49.38 years (SD = 15.826). The 95% confidence interval for the mean age was 44.25–54.51 years. Half of the 36 respondents who provided information on residential location lived in cities. Most of the 38 respondents who reported their education level had completed secondary education. The presence of chronic disease was reported by 25% of the survey participants. Detailed sociodemographic characteristics of the respondents are presented in Table 1.

### 3.2. Conflicts in Healthcare

Using a five-point Likert scale, respondents were asked to assess the effect of physician–patient communication on healthcare quality. The majority of respondents believed communication had a significant influence on healthcare quality, with 60% indicating either a strong effect (a rating of 5) or a moderate effect (a rating of 4). Approximately one-third of participants rated the effect as neither positive nor negative (rating of 3). Less than 10% of respondents rated communication as having either a minor effect (a rating of 2) or no effect at all (a rating of 1) (Figure 1).

Moreover, 80% (n = 32) of the respondents stated they wanted to receive detailed information about diseases, treatment, and medical procedures.

A significant portion of respondents reported experiencing difficulties communicating with a physician. Specifically, 60% (n = 24) indicated that they rarely encountered difficulties in communication and 5% (n = 2), reported often experiencing communication difficulties with their physician, while 35% (n = 14) stated that they never faced such challenges.

The most commonly reported communication difficulty was the limited time allocated by the physician. Insufficient information provided by the physician was identified as the second most frequent communication barrier. Additionally, a significant proportion of participants experienced challenges in understanding the information provided. Participants were able to select multiple responses for this question (Figure 2).

The questionnaire also included a question about the main reasons for conflict between physicians and patients, to which sixteen participants responded. The most frequently reported issue was that the physician had poor communication skills; this concern was cited by 31.25% (n = 5) of respondents. Another common reason was rude behavior from the physicians’ side, which was also noted by 31.25% (n = 5) of participants. A smaller proportion of respondents mentioned an inability to understand the information provided by the physician (12.50%, n = 2) and disagreement with the treatment (12.50%, n = 2) as reasons for conflicts. Only 12.50% (n = 2) identified physician incompetence as a primary cause of the conflict. Participants had the option to choose multiple responses for this question.

In addition, 95% (n = 38) of respondents claimed that conflicts cause a reduction in trust in the healthcare system, while 87.50% (n = 35) stated that conflicts have a negative impact on healthcare quality.

### 3.3. Legal System and Healthcare

Half of the participants (50% (n = 20)) stated the legal system is not effective in resolving conflicts between physicians and patients, while 45% (n = 18) could not decide on the matter. Only 5% (n = 2) viewed the legal system as effective in resolving healthcare disputes.

### 3.4. Mediation in Healthcare

Familiarity with mediation as a method for conflict resolution was limited, with less than half of the respondents reporting an awareness of it. Participation in mediation procedures was notably low, with only 10% of respondents having personal experience with mediation. When asked specifically about participating in mediation to resolve conflicts with a physician, an overwhelming majority (97.50%) reported no involvement. Similarly, nearly 90% of participants were unaware of the mediation law in Bulgaria, which has been in place since 2004, and none of the respondents were familiar with the main steps of the mediation procedure (Table 2).

The ability to preserve good relationships between the parties involved was the most commonly indicated advantage of mediation compared to filing a lawsuit, reported by 36.84% (n = 14) of respondents. Another 23.68% (n = 9) pointed to mediation’s capacity to allow for open discussion of the issue, 5.26% (n = 2) of participants viewed mediation as faster than litigation, and 65.79% (n = 25) of the respondents were uncertain about the advantages of mediation. Participants had the option to choose multiple responses for this question.

When asked whether they would use mediation services provided by healthcare institutions to resolve conflicts between physicians and patients, 45% (n = 18) of respondents expressed a willingness to use such services. An additional 35% (n = 14) indicated that they might consider using mediation and 17.50% (n = 7) could not decide. Only 2.50% (n = 1) stated they would not use mediation to resolve conflict with physician. A significant portion of respondents highlighted the importance of specialized mediation in healthcare. Specifically, 60% (n = 24) stated that if mediation was carried out by a team of mediators, including a medical professional, it would make resolving conflicts between physicians and patients easier. Additionally, 67% (n = 27) expressed that there is a need for specialized mediators in healthcare.

Several key challenges for implementing mediation in healthcare settings were identified by the participants in the survey. Building trust among physicians and patients in the mediation process was the most frequently cited challenge, which was mentioned by 40% (n = 16) of the respondents. Lack of awareness about the benefits of mediation was identified by 32.50% (n = 13) of participants as another major challenge. A quarter of respondents (25%, n = 10) stated that they could not decide on the barriers against the implementation of mediation in healthcare. The majority of respondents, 85% (n = 34), highlighted the need for increased awareness of mediation among physicians. Additionally, 87.50% (n = 35) of participants emphasized the importance of raising awareness about mediation among patients as well.

We conducted an analysis to examine potential statistically significant associations between gender and age with several key variables, including perceived impact of physician communication on healthcare quality, occurrence of conflicts between physicians and patients, the perceived effectiveness of the court system in resolving healthcare conflicts, evaluation of mediation’s potential to resolve healthcare disputes, and willingness to use mediation services provided by healthcare institutions. The Chi-Square test results revealed no statistically significant associations (*p* > 0.05). Detailed results are presented in Table 3 and Table 4.

## 4. Discussion

The majority of our respondents recognized communication as having a strong effect on healthcare quality, which is consistent with the findings of other studies. The quality of communication in healthcare plays a critical role in shaping patients’ perceptions of their illness and their attitudes toward treatment. Numerous studies highlight the link between trust and communication, with trust being a key outcome of effective communication and a major factor in patient satisfaction. A survey conducted among 778 patients hospitalized across eight public hospitals in Wroclaw reveals that sense of security correlated positively with access to information (*p* = 0.642). The most important factor influencing patient satisfaction, as reported by 82.9% of patients, was the kindness of the staff [24]. Another study identified a positive correlation between the provision of general information and patient satisfaction (r = 0.41). Patients who reported fewer informational problems tended to evaluate their hospitalization experience more positively [25]. At the same time, 65% of our respondents reported experiencing communication difficulties with their physician, which is lower than that reported in another study, in which 78% of 2086 patients in ambulatory care noted at least one interpersonal issue with their physician. In the same study, the most common reasons for communication difficulties among our respondents—lack of time allocated by the physician, insufficient information provided by the physician and problems understanding the information supplied—were found to be strongly related to trust and the overall rating of the physician [26]. These results suggest that a significant majority of participants experience challenges in communication with physicians highlight key barriers in the communication process between patients and physicians, suggesting a need for improvements in both the adequacy and clarity of information provided, as well as the time management during consultations. Furthermore, effective communication skills are essential for physicians’ daily practice, as strong communication fosters an exchange of information and actively involves patients in the decision-making process [27]. This plays an important role in ensuring adherence to prescribed treatments and medical advice, thereby enhancing the likelihood of positive health outcomes. A meta-analysis by Kelly B. Haskard Zolnierek and M. Robin DiMatteo identified a significant positive correlation between the quality of physician communication and patient adherence to medical recommendations (*p* < 0.001). Specifically, inadequate physician communication was linked to a 19% increase in the risk of patient nonadherence (r = 0.19, 95% CI = 0.16, 0.21). The robustness of these results is underscored by a fail-safe number exceeding 28,563 studies with null effects. The likelihood of nonadherence was 1.47 times higher when physician communication was poor, whereas the odds of adherence improved by 2.16 times when communication was effective [28]. Another study involving 2000 patients showed a significant positive association between provider communication and overall self-management (β = 0.10, *p* < 0.001, model R^2^ = 0.21). Additionally, provider communication was found to be strongly linked to key aspects of self-care [29]. Better communication benefits not only patients but also physicians, as it enhances job satisfaction, reduces work-related stress, and lowers the risk of malpractice litigation and professional burnout [30,31,32].

Moreover, the findings of our study suggest that the primary sources of conflict in healthcare are more closely related to communication barriers and interpersonal issues, which were reported as the main sources of conflict by most of our respondents rather than physician competence, which was indicated by only 12.50%. These findings align with those from another study of complaints from an academic medical center, where the most common reason for complaining was challenges with communication [4]. This is further supported by the fact that the most common reason for filing a lawsuit against a physician is not medical error, but rather a breakdown in the patient–physician relationship, primarily characterized by unsatisfactory communication between patients and physicians [33]. This is confirmed by another study—poor communication with the physician was cited as a key reason for filing a lawsuit, with common complaints including a lack of information about potential health issues (70%), feeling misled (48%), unanswered questions (32%), and ignored concerns (13%). Similarly, a review of plaintiff depositions showed that 71% of cases involved communication failures, such as not listening to patients or providing clear explanations about adverse events. The authors even suggest that communication failures were more prevalent than originally recognized. A survey by Charles Vincent and colleagues which involved over 200 patients engaged in malpractice litigation also highlighted dissatisfaction with post-incident communication [34].

The legal system, often regarded as the primary method for resolving conflicts between patients and physicians, was reported to be effective by only 5% of respondents in our survey. The majority (95%) expressed skepticism or uncertainty about its effectiveness. This is consistent with Hambali’s observations, which defined the legal system as ineffective because of its high costs, lengthy timeframes for verdicts, and impractical complexity [11]. In another study, Dr. Puteri Nemie and Jahn Kassim state in that the court system often fails to deliver a fair verdict due to shortcomings such as unequal legal representation, imperfections in evidence, unreliable witnesses, and subjectivity [35]. The plaintiff must prove the defendant’s fault, and the judge rules based on evidence, witness testimonies, expert opinions, and the law. Strict procedures and rules make things difficult for plaintiffs, especially patients [36]. Furthermore, expert witnesses are crucial in healthcare cases, but many physicians refuse to serve as one due to heavy workloads, delays, time constraints, and unfamiliarity with court procedures [37]. As a result, many victims of medical errors do not receive compensation, and even when awarded, the amount is often unpredictable. Additionally, patients may have to pay attorneys up to 30–40% of the compensation. Healthcare litigation is also costly: legal expenses in the U.S. exceed USD 55 billion annually, while legal disputes in healthcare in the United Kingdom account for 0.04% of its gross domestic product [9,35,38,39]. Furthermore, high administrative and legal costs are a barrier for many patients to file a lawsuit. In Canada, the cost of initiating a lawsuit can reach between CAD 40,000 and CAD 50,000 [40]. Mediation, on the other hand, is cheaper, requires less preparation, and often eliminates the need for lawyers [41]. Another significant disadvantage of the traditional judicial system is the lengthy time required to make a ruling and conclude a case. According to Jury Verdict Research, the average amount of time taken to resolve a medical malpractice case in the United States in 2000 was 45 months. In Malaysia, the duration of a medical malpractice trial can take a minimum of 15 years and may extend up to 25 years. In Canada, the time that passes between the initial filing of a claim and its appearance in court is typically 5 to 6 years. In contrast, a study that included 13 mediation firms in the U.S. found that the average time taken to reach a resolution through mediation is just 1 to 3 days [11,40,42].

Maintaining positive relationships between parties and the ability to foster open dialogue were recognized by our study participants as key advantages of mediation compared to the judicial system. This perspective aligns with the observations of Gulton, an American mediator with expertise in medical negligence cases, who has overseen more than 600 disputes. Gulton emphasizes that mediation offers injured plaintiffs and their families a “therapeutic resolution,” extending beyond financial compensation to include explanations, apologies, expressions of regret, empathy, and sympathy, along with opportunities for closure, forgiveness, and the rebuilding of valued relationships [43]. This results in a 90% satisfaction rate for both parties [42].

The majority of our respondents indicated that the traditional court system is ineffective in resolving conflicts between physicians and patients, and mediation may offer a more efficient conflict resolution method by reducing time, costs, and enhancing satisfaction among the parties involved [44]. Familiarity with mediation was reported by less than half of the respondents, only 10% had participated in mediation, and just 2.50% had participated in a mediation procedure with a physician. Additionally, only 12.50% of participants were aware that a law on mediation was established in Bulgaria in 2004 [21] and none of the respondents were familiar with the main steps of the mediation procedure. The key challenges identified by the participants which prevented the implementation of mediation in healthcare included building trust between physicians and patients and lack of awareness about the benefits of mediation. Additionally, more than 80% of respondents highlighted the need for increased awareness of mediation among physicians and patients. At the same time, 80% of respondents expressed willingness or consideration to utilize mediation services in healthcare conflicts if they were available. These findings suggest that the majority of respondents are unfamiliar with mediation but express a willingness to use it for conflict resolution in healthcare settings. This underscores the need for deliberate efforts to build trust among physicians and patients, as trust is a critical foundation for the successful implementation of mediation. Equally important is the development of comprehensive, targeted awareness initiatives aimed at educating both healthcare professionals and patients about the process, benefits, and availability of mediation services. These efforts are essential for enhancing the acceptance, utilization, and overall effectiveness of mediation as a conflict resolution method in healthcare settings.

The statistical power of this study’s findings is limited by the relatively small sample size, consisting of only 40 respondents. This limitation may be the reason for the lack of statistically significant correlations between the different variables examined and may reduce the generalizability of the results.

Another limitation of the study is the geographic scope, as only Bulgarian citizens were included in this study. Although the findings provide important insights into the feasibility of mediation in the Bulgarian context, caution should be exercised when generalizing the results to other geographical regions. The applicability of the conclusions to populations in other countries may be influenced by different cultural, social, legal and healthcare system factors.

The success of mediation in healthcare is likely to be influenced by the other key stakeholders, including mediators and physicians. Opinions regarding the benefits, challenges, and overall feasibility of mediation from these groups are critical to developing a more comprehensive understanding of how mediation can be integrated into healthcare settings.

As this pilot study reveals the interest of one of the key stakeholder groups in the topic, it serves as a foundation for future research. A crucial next step is to address the identified limitations by substantially increasing the sample size (which will include incorporating different geographical regions and more stakeholders). This will support the development of a more comprehensive understanding of the feasibility and effectiveness of mediation in various healthcare settings.

## 5. Conclusions

This study highlights the potential of mediation as a valuable alternative to litigation in resolving healthcare disputes. While communication barriers between physicians and patients are a significant concern, with many patients unaware of mediation options (60% in this study), the results underscore the importance of raising awareness about mediation as a means to improve healthcare outcomes. The lack of trust and limited knowledge of mediation are significant obstacles that hinder its wider adoption. Future efforts should prioritize educating both healthcare professionals and patients about the benefits of mediation, integrating it into healthcare systems to foster better conflict resolution and strengthen trust in the healthcare system.

## Figures and Tables

**Figure 1 healthcare-13-00629-f001:**
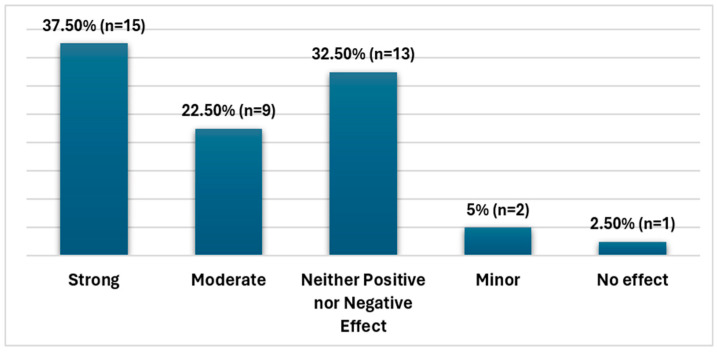
Perceived effect of communication on healthcare quality.

**Figure 2 healthcare-13-00629-f002:**
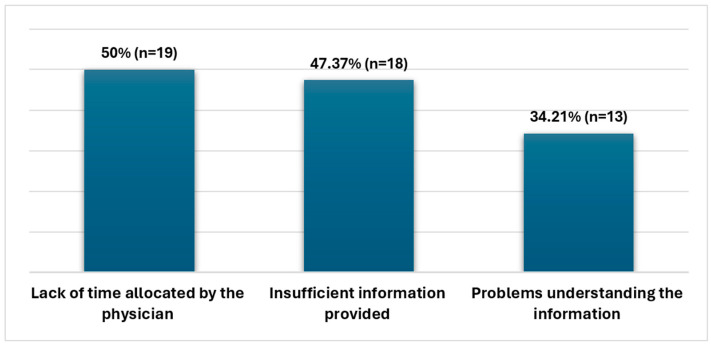
Reasons for communication difficulties with physicians.

**Table 1 healthcare-13-00629-t001:** Demographic characteristics of the respondents.

Indicator	Category	N	Relative Proportion (%)	Standard Error of the Proportion (SEp)
Gender	Male	16	40	±7.75
	Female	24	60	±7.75
Residency	City	18	50	±8.33
	Town	11	30.56	±7.68
	Village	7	19.44	±6.60
Education	Primary	1	2.63	±2.60
	Secondary	21	55.26	±8.10
	Higher	16	42.11	±8.00
Presence of chronic disease	Yes	10	25	±6.85
	No	30	75	±6.85

**Table 2 healthcare-13-00629-t002:** Awareness of and participation in mediation procedure for conflict resolution.

Category/Question	Response	Percentage (%)	Count (n)
Familiarity with mediation as a method of conflict resolution	Yes	40%	16
	No	60%	24
Participation in any mediation procedure	Yes	10%	4
	No	90%	36
Participation in mediation to resolve a conflict with a physician	Yes	2.50%	1
	No	97.50%	39
Awareness of the mediation law in Bulgaria (since 2004)	Yes	12.50%	5
	No	87.50%	35

**Table 3 healthcare-13-00629-t003:** Chi-Square analysis of gender-based associations in healthcare communication, conflict, court system effectiveness, and mediation perceptions.

Variable Paired with Gender	Male%	Female%	Chi-Square Value (χ^2^)	*p*-Value
**Perceived Impact of Physician Communication on Healthcare Quality (Scale 1 to 5)**			5.057	0.282
1 (No effect)	6.25	0		
2	0	8.33		
3	43.75	25		
4	25	20.83		
5 (Strong effect)	25	45.83		
**Conflict with a Physician**			3.259	0.071
Yes	31.25	8.70		
No	68.75	91.30		
**Court System Assessment in Healthcare Conflicts**			1.551	0.460
Effective	0	8.33		
Ineffective	56.25	45.83		
I cannot decide	43.75	45.83		
**Assessment of Mediation’s Potential in Healthcare Conflicts**			5.519	0.137
No	18.75	4.17		
Yes, to a limited extent	43.75	25		
Yes, to a large extent	6.25	25		
I cannot decide	31.25	45.83		
**Willingness to Use Healthcare-Provided Mediator Services**			2.136	0.545
Yes	50	41.67		
No	6.25	0		
Maybe	31.25	37.50		
I cannot decide	12.50	20.83		

**Table 4 healthcare-13-00629-t004:** Chi-Square analysis of age-based associations in healthcare communication, conflict, court system effectiveness, and mediation perceptions.

Variable Paired with Age Group	Years% ≤45	Years% 46–60	Years% ≥61	Chi-Square Value (χ^2^)	*p*-Value
**Perceived Impact of Physician Communication on Healthcare Quality (Scale 1 to 5)**				8.958	0.346
1 (No effect)	0	10	0		
2	5.26	0	9.09		
3	47.37	10	27.27		
4	21.05	20	27.27		
5 (Strong effect)	26.32	60	36.36		
**Conflict with a Physician**				1.592	0.451
Yes	16.67	30	9.09		
No	83.33	70	90.91		
**Court System Assessment in Healthcare Conflicts**				2.374	0.667
Effective	10.53	0	0		
Ineffective	47.37	50	54.55		
I Cannot Decide	42.11	50	45.45		
**Assessment of Mediation’s Potential in Healthcare Conflicts**				3.892	0.691
No	15.79	10	0		
Yes, to a Limited Extent	26.32	40	36.36		
Yes, to a Large Extent	10.53	20	27.27		
I Cannot Decide	47.37	30	36.36		
**Willingness to Use Healthcare-Provided Mediator Services**				5.279	0.509
Yes	42.11	30	63.64		
No	0	10	0		
Maybe	36.84	40	27.27		
I Cannot Decide	21.05	20	9.09		

## Data Availability

Data will be provided on request.

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
