# Peer review of "Exploring Patient Awareness and the Feasibility of Mediation in Healthcare: A Pilot Study in Bulgaria"

_healthcare, 2025, doi:10.3390/healthcare13060629_

Round 1
Reviewer 1 Report
Comments and Suggestions for Authors
1. In line 30, replace "it is personal" with "the process of providing and receiving needed healthcare services is personal" for purposes of clarification.
2. It is not clear in the methods section whether the questionnaire developed for the purpose of the study used validated questions. In case no validated questionnaires were utilized to measure items related to patient-provider communication and sociodemographic characteristics, is there a reason why? Either way, this needs to be clarified in the methods.
3. Lines 93-97: Rephrase the statistical methods used in paragraph format since it is confusing to the reader when trying to read the different statistical methods employed. It is also not clear which associations were tested using chi-squared test. Also, why chi-squared compared to fisher's exact test when you have a small sample size and small cell sizes across variables? I believe fisher's exact test is more relevant in your case.
4. In the results, remove the decimal points when reporting full numbers, so 40% instead of 40.00%. Also, please have a separate paragraph to share all the results for the fisher's exact test carried out.
5. lines 204-208 belong in the results not in the discussion. Please delete.
6. In line 244, in-text citation is incorrect. Please correct the reference to the authors.
7. This study has several limitations from sample size, to generalizability issues, to non-validated questionnaire items. A limitations section is required in the discussion. Please add one.
Author Response
Dear Reviewer,
We would like to sincerely thank you for your thoughtful and insightful recommendations. We have carefully addressed each of your points and have made the necessary revisions. We trust that these adjustments will align with your expectations and enhance the quality of the work. Your feedback has been immensely helpful, and we are grateful for your time and expertise in reviewing our submission.
Comments 1: In line 30, replace "it is personal" with "the process of providing and receiving needed healthcare services is personal" for purposes of clarification.
Response1: We have revised the sentence on line 30 as follows:
Furthermore, the process of providing and receiving needed healthcare services is personal and frequently provokes intense emotions in both the patient and healthcare professionals [1].
Comments 2: It is not clear in the methods section whether the questionnaire developed for the purpose of the study used validated questions. In case no validated questionnaires were utilized to measure items related to patient-provider communication and sociodemographic characteristics, is there a reason why? Either way, this needs to be clarified in the methods.
Response 2: Dear Reviewer, in response to your recommendation, we have now included the development process of the questionnaire in the Methods section for greater clarity.We were unable to find an existing validated questionnaire on patient-provider communication that would suit the specific aims of our study. As detailed in the Results section, the questionnaire was developed by the first author, a certified mediator and medical doctor, based on a comprehensive review of the relevant literature. To ensure that the tool was well-aligned with the study’s objectives and accurately captured the intended constructs, it was subsequently validated by the second author, a certified mediator and dental doctor. This validation process focused on assessing the content relevance, clarity, and reliability of the questionnaire in measuring the key aspects of the study.
The Materials and methods section was revised as follows lines 102-108:
The questionnaire was created by the first author, a certified mediator and medical doctor, following a comprehensive literature review. To ensure the tool's alignment with the study's objectives and its ability to accurately measure the intended constructs, it was validated by the second author, a certified mediator and dental doctor. The validation process focused on ensuring the questionnaire’s content relevance, clarity, and reliability in capturing the key aspects of the research.
Comments 3: Lines 93-97: Rephrase the statistical methods used in paragraph format since it is confusing to the reader when trying to read the different statistical methods employed. It is also not clear which associations were tested using chi-squared test. Also, why chi-squared compared to fisher's exact test when you have a small sample size and small cell sizes across variables? I believe fisher's exact test is more relevant in your case.
Response 3: The statistical methods employed are outlined in lines 119-135 and have been rephrased in or improved clarity and readability. These methods are also described in greater detail, as suggested by other reviewers. The associations tested are now included in the Materials and Methods section. Additionally, the detailed results of the chi-square test are presented in Table 3 and Table 4, starting from line 244.
Forty participants completed the questionnaire. The data were analyzed using IBM SPSS Statistics v. 19, employing the following statistical methods:
- Descriptive Statistics: Used to summarize and describe the main features of the collected data. This included calculating frequency counts and percentages for categorical variables, as well as measures of central tendency (mean) and dispersion (standard deviation) for continuous variables. Additionally, confidence intervals (95% CI) were calculated for the mean to provide an estimate of the range within which the true population mean is likely to fall.
- Shapiro-Wilk test: Used to assess whether the age variable was normally distributed. The test was performed at a significance level of p = 0.05.
- Pearson’s Chi-Square test: Applied to evaluate associations between gender and age with the following variables: perceived impact of physician communication on healthcare quality, occurrence of communication difficulties and conflicts between physician and patient, perceived effectiveness of the court system in resolving healthcare conflicts, evaluation of mediation's potential to resolve healthcare disputes, and willingness to use mediation services provided by healthcare institutions. The significance level chosen for the analysis was set at p = 0.05.
Thank you for your valuable comment regarding the choice between Pearson’s Chi-Square test and Fisher’s Exact Test, particularly given the small sample size and cell frequencies. We chose to use Pearson’s Chi-Square test because, despite the sample size being relatively small, it includes more than 30 participants, which is generally considered acceptable for the Chi-Square approximation. Additionally, upon examining the expected frequencies in our contingency table, we found that the majority of cells met the threshold of ≥ 5, which aligns with the typical assumption for using Pearson’s Chi-Square test. Furthermore, since all of our contingency tables were larger than 2x2, this further aligned with the rationale for using the Chi-Square test in SPSS. We believe this approach is appropriate given the characteristics of our data.
Comments 4: In the results, remove the decimal points when reporting full numbers, so 40% instead of 40.00%. Also, please have a separate paragraph to share all the results for the fisher's exact test carried out.
Response 4: We have removed the decimal points from all full numbers reported, including those in the figures, as suggested.
Comments 5: lines 204-208 belong in the results not in the discussion. Please delete.
Response5: We have moved the content from suggested lines. The section has been updated accordingly and now begins as follows (lines 252-253 in the revised manuscript):
The majority of our respondents recognized communication as having strong effect on healthcare quality, which is consistent with the findings of other studies.
Comments 6: In line 244, in-text citation is incorrect. Please correct the reference to the authors.
Response 6: Dear Reviewer, we do not believe there are any mistakes in the citation. Reference 11 (line 316 in the revised manuscript), cited in the sentence, corresponds to the statement regarding Hambali's study, which states: "The majority (95%) expressed skepticism or uncertainty about its effectiveness. This is consistent with Hambali's observations, which defined the legal system as ineffective because of its high costs, lengthy timeframes for verdicts, and impractical complexity [11]." The full reference is as follows:
- Hambali SN, Khodapanahandeh S. A Review of Medical Malpractice Issues in Malaysia under the Tort Litigation System. Global Journal of Health Science. 2014 Apr 7;6(4). https://doi.org/10.5539/gjhs.v6n4p76.
To clarify the distinction between this statement and the following one, where the misunderstanding may have originated, we have revised the beginning of the subsequent sentence (line 316):
In another study, Dr. Puteri Nemie and Jahn Kassim state in their article that the court system often fails to deliver a fair verdict due to shortcomings such as unequal legal representation, imperfections in evidence, unreliable witnesses, and subjectivity [25].
Comments 7: This study has several limitations from sample size, to generalizability issues, to non-validated questionnaire items. A limitations section is required in the discussion. Please add one.
Response 7: A paragraph discussing the limitations of the study in detail has been added at the end of the discussion section (lines 370-390).
The statistical power of this study’s findings is limited by the relatively small sample size, consisting of only 40 respondents. This limitation may be the reason for the lack of statistically significant correlations between the different variables examined and may reduce the generalizability of the results.
Another limitation of the study is the geographic scope, as participants included were only Bulgarian citizens. Although the findings provide important insights into the feasibility of mediation in the Bulgarian context, the results should be generalized with caution to other geographical regions. The applicability of the conclusions to populations in other countries may be influenced by different cultural, social, legal and healthcare system factors.
Тhe success of mediation in healthcare is likely to be influenced by the other key stakeholders, including mediators and physicians. Opinions regarding the benefits, challenges, and overall feasibility of mediation from these groups are critical to developing a more comprehensive understanding of how mediation can be integrated into healthcare settings.
As this pilot study reveals the interest of one of the key stakeholder groups in the topic, it serves as a foundation for future research. A crucial next step is to address the identified limitations by a substantial increase in the sample size (including different geographical regions and more stakeholders). This will support the development of a more comprehensive understanding of the feasibility and effectiveness of mediation in various healthcare settings.
Reviewer 2 Report
Comments and Suggestions for Authors
I read with interest the pilot study presented by Dimitrov and Miteva-Katrandzhieva on patients awareness and feasibility of mediation in the healthcare setting.
The study presents a novel, yet essential aspect of healthcare services, which is mediation as an alternative conflict resolution method in healthcare.
The study is fits the scope of the journal, and is of interest to the readership of journal.
However, I cannot recommend the article to be accepted in its current state, as it is riddled with many major issues that require the authors' attention:
1- Although the study title covers the important aspects of the study, the study presented a very small population in Bulgaria. While it may be argued that the sample size is fit for a pilot study, I believe authors should state the locality of the study in the title, as its finding cannot be generalised to other populations.
2- While I appreciate the extensive introduction, authors should dedicate a separate paragraph for the gap of literature which prompted the authors to carry out this study. The paragraph should also cover the aims and objectives of this study.
3- It is unclear why the authors resorted to a pilot study, rather than a full scale study.
4- Materials and Methods should be explained explicitly to facilitate the study transparency and reproducibility, as explained in the subsequent points.
5- The study tool, i.e., the questionnaire, should be further described in the Materials and Methods section. How was it developed? how was it validated? in which language was it administered? Was the survey conducted in-person or online?
6- A copy of the questionnaire, or at least a translated copy of it in English should be included in the study.
7- How did the authors determine the sample size?
8- Authors should further expand on their statistical analysis section. For example, what were the descriptive statistics used?
9- In the results section, authors should summarize the findings presented in the Table, rather describing the entire table in writing.
10- How did the authors determine the presence of chronic disease? What was their definition of chronic diseases?
11- Authors are strongly encouraged to revise their naming of the effect in their scale, as it is inaccurate, and sometimes incorrect. For example, a score of 3 was referred to as "Neutral" which could indicate no effect, while a score of 1 was reported as "no effect".
12- While I encourage presenting the study findings in different formats, such as tables, graphs and charts, I find the results presentation in this study is ineffective and does not contribute significantly to the article readership. For example, figures 1 and 2 are redundant and already described in the text.
13- Authors are encouraged to present their analytical findings, e.g., association between participants' characteristics and the various matters (perceived effect of communication, origins of communications difficulties ...etc) in a table format, along with the P values.
14- The discussion section should be referenced appropriately and discuss the study findings in light of the current literature. It was quite surprising to see that the introduction was very well presented with 17 references, while the discussion contained only 12.
15- Authors should dedicate an entire paragraph at the end of their discussion to discuss the limitations of their current study in depth.
16- The conclusion of the study should be shortened to better reflect what is concluded from this study.
Author Response
Dear Reviewer,
We would like to sincerely thank you for your thoughtful and insightful recommendations. We have carefully addressed each of your points and have made the necessary revisions. We trust that these adjustments will align with your expectations and enhance the quality of the work. Your feedback has been immensely helpful, and we are grateful for your time and expertise in reviewing our submission.
Comments 1: Although the study title covers the important aspects of the study, the study presented a very small population in Bulgaria. While it may be argued that the sample size is fit for a pilot study, I believe authors should state the locality of the study in the title, as its finding cannot be generalised to other populations.
Response 1: The title of the study has been revised as follows:
“Exploring Patient Awareness and Feasibility of Mediation in Healthcare: A Pilot Study in Bulgaria”
Comments 2: While I appreciate the extensive introduction, authors should dedicate a separate paragraph for the gap of literature which prompted the authors to carry out this study. The paragraph should also cover the aims and objectives of this study.
Response 2: As per your recommendation, we have created a new paragraph to address the gap in the literature that prompted us to carry out this study (lines 81-90). The paragraph also outlines the aims and objectives of the study, as follows:
The main goal of this study is to evaluate patients' awareness of mediation and to investigate the feasibility of introducing mediation as an alternative to the court system for resolving disputes within the healthcare system. At the time of this study, no similar research had been conducted in Bulgaria. Furthermore, the available literature does not provide insights into how mediation could be effectively incorporated into the Bulgarian healthcare system. Given the need for exploring alternative dispute resolution methods to address healthcare disputes in the country, along with the existing legislation on mediation, the gap in the literature has driven the need for this study, highlighting its importance, as mediation is still not widely practiced or studied for healthcare dispute resolution in Bulgaria.
Comments 3: It is unclear why the authors resorted to a pilot study, rather than a full-scale study.
Response 3: The reason we chose to conduct a pilot study rather than a full-scale study is that the questionnaire is part of a larger PhD thesis project. The results of the full study will be published in separate articles once the data collection and analysis are completed.
Comments 4: Materials and Methods should be explained explicitly to facilitate the study transparency and reproducibility, as explained in the subsequent points.
Response 4: Thank you for your valuable feedback. In response to your comment, we have addressed the subsequent points to provide a clearer and more detailed explanation of the Materials and Methods section.
Comments 5: The study tool, i.e., the questionnaire, should be further described in the Materials and Methods section. How was it developed? how was it validated? in which language was it administered? Was the survey conducted in-person or online?
Response 5: Materials and Methods section has been revised, and the recommended details have been incorporated as follows:
Lines 92-94: A pilot cross-sectional study was conducted using an anonymous survey. The survey was administered online using the Lime Survey platform between July 1, 2024, and August 1, 2024.
Lines 102-108: For the study, a structured questionnaire in Bulgarian language was developed. The questionnaire was created by the first author, a certified mediator and medical doctor, following a comprehensive literature review. To ensure the tool's alignment with the study's objectives and its ability to accurately measure the intended constructs, it was validated by the second author, a certified mediator and dental doctor. The validation process focused on ensuring the questionnaire’s content relevance, clarity, and reliability in capturing the key aspects of the research.
Comments 6: A copy of the questionnaire, or at least a translated copy of it in English should be included in the study.
Response 6: The questionnaire was generated as part of a larger PhD thesis project. As such, the final results and the complete questionnaire will be published in a separate article. We appreciate your understanding.
Comments 7: How did the authors determine the sample size?
Response 7: Dear Reviewer, thank you for your comment. According to the literature, the median sample size for a pilot study is typically around 30 participants (please refer to the links below). In our study, we chose to include at least 40 participants to assess whether the residents of Bulgaria are open to mediation as an alternative to the court system for conflict resolution and to evaluate if lack of awareness is a key obstacle to the wider implementation of mediation in healthcare or if there are other underlying issues. This sample size allowed us to gain preliminary insights and plan for a full-scale study, the results of which will be published in the future.
A review of sample sizes for UK pilot and feasibility studies on the ISRCTN registry from 2013 to 2020 | Pilot and Feasibility Studies | Full Text
A simple formula for the calculation of sample size in pilot studies - ScienceDirect
Comments 8: Authors should further expand on their statistical analysis section. For example, what were the descriptive statistics used?
Response 8: As per your suggestion, the statistical analysis section has been expanded. It was paraphrased for better readability, as recommended by another reviewer. The revised version is provided below (lines 119-135):
Forty participants completed the questionnaire. The data were analyzed using IBM SPSS Statistics v. 19, employing the following statistical methods:
- Descriptive Statistics: Used to summarize and describe the main features of the collected data. This included calculating frequency counts and percentages for categorical variables, as well as measures of central tendency (mean) and dispersion (standard deviation) for continuous variables. Additionally, confidence intervals (95% CI) were calculated for the mean to provide an estimate of the range within which the true population mean is likely to fall.
- Shapiro-Wilk test: Used to assess whether the age variable was normally distributed. The test was performed at a significance level of p = 0.05.
- Pearson’s Chi-Square test: Applied to evaluate associations between gender and age with the following variables: perceived impact of physician communication on healthcare quality, occurrence of communication difficulties and conflicts between physician and patient, perceived effectiveness of the court system in resolving healthcare conflicts, evaluation of mediation's potential to resolve healthcare disputes, and willingness to use mediation services provided by healthcare institutions. The significance level chosen for the analysis was set at p = 0.05.
Comments 9: In the results section, authors should summarize the findings presented in the Table, rather describing the entire table in writing.
Response 9: Dear Reviewer, as recommended, we have revised the results section by summarizing the sociodemographic findings in the text and referring to the table for detailed information lines (lines 138-145)
Of the total 40 respondents, women were more prevalent. The Shapiro-Wilk test indicated that the age variable was normally distributed (p = 0.230, DF = 39). Participants’ ages ranged from 21 to 80 years, with a mean age of 49.38 years (SD = 15.826). The 95% confidence interval for the mean age was 44.25 - 54.51 years. Half of the 36 respondents who provided information on residential location lived in cities. Most of the 38 respondents who reported their education level completed secondary education. The presence of chronic disease was reported by 25% of the survey participants. Detailed sociodemographic characteristics of the respondents are presented in Table 1.
Comments 10: How did the authors determine the presence of chronic disease? What was their definition of chronic diseases?
Response 10: Chronic disease was defined in the study as a condition affecting a specific organ or system of organs, typically characterized by acute onset followed by recurrent episodes of varying duration, with symptoms persisting or reappearing for a period exceeding one year. The questionnaire used in the study included a direct question regarding the presence of a chronic disease. Respondents were asked to self-assess and indicate whether they had a chronic condition, reflecting their personal evaluation of their health status.
Comments 11: Authors are strongly encouraged to revise their naming of the effect in their scale, as it is inaccurate, and sometimes incorrect. For example, a score of 3 was referred to as "Neutral" which could indicate no effect, while a score of 1 was reported as "no effect".
Response 11: Dear Reviewer, the naming of the effect for the scale scores has been revised to ensure accuracy and clarity. Specifically, the term "Neutral" has been refined to indicate "Neither Positive Effect nor Negative Effect" for a score of 3 in line 155. Furthermore, Figure 1, "Perceived Effect of Communication on Healthcare Quality," has been revised accordingly to reflect these changes.
Comments 12: While I encourage presenting the study findings in different formats, such as tables, graphs and charts, I find the results presentation in this study is ineffective and does not contribute significantly to the article readership. For example, figures 1 and 2 are redundant and already described in the text.
Response 12: Thank you for your valuable feedback. In response to your comments, we have revised the figures (lines 157 and 171) to ensure that all data percentages and numbers are clearly presented within the visuals, thereby eliminating redundancy with the text. We have also adjusted the text to avoid any overlap (lines 151 -157 and lines 167-171). Additionally, we have included three additional tables detailing the results of the chi-square test (line 245 and 248), as well as the familiarity and participation in mediation (line 205), which we believe will further enhance the readability and clarity of the study’s findings.
Comments 13: Authors are encouraged to present their analytical findings, e.g., association between participants' characteristics and the various matters (perceived effect of communication, origins of communications difficulties ...etc) in a table format, along with the P values.
Response 13: Thank you for your comment. We have included two additional tables that present the detailed results from the chi-square test and the associations between the tested variables and age or gender (lines 245 and 248).
Comments 14: The discussion section should be referenced appropriately and discuss the study findings in light of the current literature. It was quite surprising to see that the introduction was very well presented with 17 references, while the discussion contained only 12.
Response 14: The discussion section has been revised to provide a more comprehensive analysis, with additional details incorporated in lines 247-292, 301-310, and 306-326. We have also added more references.
Comments 15: Authors should dedicate an entire paragraph at the end of their discussion to discuss the limitations of their current study in depth.
Response 15: A paragraph discussing the limitations of the study in detail has been added at the end of the discussion section (lines 370-390).
The statistical power of this study’s findings is limited by the relatively small sample size, consisting of only 40 respondents. This limitation may be the reason for the lack of statistically significant correlations between the different variables examined and may reduce the generalizability of the results.
Another limitation of the study is the geographic scope, as participants included were only Bulgarian citizens. Although the findings provide important insights into the feasibility of mediation in the Bulgarian context, the results should be generalized with caution to other geographical regions. The applicability of the conclusions to populations in other countries may be influenced by different cultural, social, legal and healthcare system factors.
Тhe success of mediation in healthcare is likely to be influenced by the other key stakeholders, including mediators and physicians. Opinions regarding the benefits, challenges, and overall feasibility of mediation from these groups are critical to developing a more comprehensive understanding of how mediation can be integrated into healthcare settings.
As this pilot study reveals the interest of one of the key stakeholder groups in the topic, it serves as a foundation for future research. A crucial next step is to address the identified limitations by a substantial increase in the sample size (including different geographical regions and more stakeholders). This will support the development of a more comprehensive understanding of the feasibility and effectiveness of mediation in various healthcare settings.
Comments 16: The conclusion of the study should be shortened to better reflect what is concluded from this study.
Response 16: The conclusion of the study has been revised as recommended and is now presented in lines 292-400.
This study highlights the potential of mediation as a valuable alternative to litigation in resolving healthcare disputes. While communication barriers between physicians and patients are a significant concern, with many patients unaware of mediation options (60% in this study), the results underscore the importance of raising awareness about mediation as a means to improve healthcare outcomes. The lack of trust and limited knowledge of mediation are significant obstacles that hinder its wider adoption. Future efforts should prioritize educating both healthcare professionals and patients about the benefits of mediation, integrating it into healthcare systems to foster better conflict resolution and strengthen trust in the healthcare system.
Reviewer 3 Report
Comments and Suggestions for Authors
The authors have investigated the informative burden, the role of communication and then the awareness of mediation procedure in Bulgaria on a 40 citizens sample.
The issues regarding information disclosure, informed consent and shared decision-making, and alternative conflict resolution in healthcare, even weighed by the sample size, are consistent with the journal aim and interesting.
abstract: please substitute residents with citizens
introduction: properly set the background. Please state for cross-sectional comparisons if alternative conflict resolution are compulsory before trial claiming or optional.
m&m line 80 universal coverage and mandatory prophylactic check-ups. I do not understand the sentence's role on the study.
line 91 specify if answering all 44 questions is mandatory or optional
results: lines 114-115 and 150-151 please move to methods after questionnaire description as more consistent with the methodology.
lines 135-140 very relevant. To be further discussed and references in the discussion regarding the role of information disclosure and patient comprehension for shared decision-making in healthcare, also associated with litigation risks. Useful also global comparisons on information and communication for consenting.
3.4 probably, as 3.2, it could be useful to add tables to increase readability.
discussion: see previous suggestion.
line 246 reference 25 extremely limited. Juridical systems are challenged with some limitations but still represent the proper and adequate mechanism and inequality could not be questioned for the trial system and not similarly to mediation. Please weigh and discuss further, with further referencing.
conclusions: to be weighed as the article seems more focused on information and communication issues while are limited those on mediation (60% are unaware of mediation)
institutional review board: please remove sentence on anonymization, as difficult to sustain and irrelevant.
informed consent: rephrase as informed consent is linked with participation in the research, even sociological, and then for data management.
many thanks
Author Response
Dear Reviewer,
We would like to sincerely thank you for your thoughtful and insightful recommendations. We have carefully addressed each of your points and have made the necessary revisions. We trust that these adjustments will align with your expectations and enhance the quality of the work. Your feedback has been immensely helpful, and we are grateful for your time and expertise in reviewing our submission.
Comments 1: abstract: please substitute residents with citizens
Response 1: As suggested, the term "residents" in line 14 has been replaced with "citizens"
Comments 2: introduction: properly set the background. Please state for cross-sectional comparisons if alternative conflict resolution are compulsory before trial claiming or optional.
Response 2: Dear Reviewer, while most countries have legislation related to mediation, we found no evidence of any country implementing a mandatory mediation procedure specifically for healthcare conflict resolution prior to trial. However, some countries do offer voluntary mediation services and have gained experience in resolving healthcare disputes through such systems. To address this point, we have added the following paragraph in lines 72-80:
Bulgaria introduced a law on mediation in 2004, and following the adoption of the European Parliament's Mediation Directive 2008/52/EC, all EU member states were required to incorporate the directive into their legal systems by 21 May 2011 [21,22]. However, mediation remains a relatively new and underutilized approach in healthcare, particularly in Bulgaria. In contrast, countries like the UK have successfully integrated mediation into their healthcare systems. After a promising pilot program, the UK’s National Health Service formally launched mediation services for claims in December 2016. Between its launch and March 2019, 606 healthcare cases were handled, with 74% resolved within 28 days [23].
Comments 3: m&m line 80 universal coverage and mandatory prophylactic check-ups. I do not understand the sentence's role on the study.
Response 3: Thank you for your comment. The sentence in line 80, “This broader approach was chosen due to the universal nature of healthcare interaction in Bulgaria, where mandatory prophylactic medical check-ups ensure that virtually all citizens have contact with medical establishments.” has been rephrased for better clarity and readability as follows (in line 97 in the revised manuscript):
This broader approach was selected because Bulgarian legislation requires annual prophylactic medical check-ups, which ensures that all respondents have had frequent interactions with the healthcare system shaping their own opinion over the topics included in the survey.
Comments 4: line 91 specify if answering all 44 questions is mandatory or optiona
Response 4: Thank you for your comment. All questions in the survey were optional.
Comments 5: results: lines 114-115 and 150-151 please move to methods after questionnaire description as more consistent with the methodology.
Response 5: The sentence, “The study employed a five-point Likert scale from 1 (no effect) to 5 (strong effect) to examine the effect of physician-patient communication on perceived healthcare quality,” from lines 114-115 has been relocated to lines 112-114 in the Materials and Methods section.
Comment 6: lines 135-140 very relevant. To be further discussed and references in the discussion regarding the role of information disclosure and patient comprehension for shared decision-making in healthcare, also associated with litigation risks. Useful also global comparisons on information and communication for consenting.
Response 6: The recommended topics have been further discussed. 274-292 focusing on the role of information disclosure and patient comprehension in shared decision-making in healthcare.
Furthermore, effective communication skills are essential for physicians’ daily practice, as strong communication fosters the exchange of information and actively involves patients in the decision-making process [27]. This plays an important role in ensuring adherence to prescribed treatments and medical advice, thereby enhancing the likelihood of positive health outcomes. A meta-analysis by Kelly B. Haskard Zolnierek and M. Robin DiMatteo identified a significant positive correlation between the quality of physician communication and patient adherence to medical recommendations (P < 0.001). Specifically, inadequate physician communication was linked to a 19% increase in the risk of patient nonadherence (r = 0.19, 95% CI = 0.16, 0.21). The robustness of these results is underscored by a fail-safe number exceeding 28,563 studies with null effects. The likelihood of nonadherence was 1.47 times higher when physician communication was poor, whereas the odds of adherence improved by 2.16 times when communication was effective [28]. Another study involving 2,000 patients showed a significant positive association between provider communication and overall self-management (β = 0.10, P < .001, model R² = .21). Additionally, provider communication was found to be strongly linked to key aspects of self-care [29]. Better communication benefits not only patients but also physicians, as it enhances job satisfaction, reduces work-related stress, and lowers the risk of malpractice litigation and professional burnout [30-32].
Additionally, the association between poor communication and litigation risks has been addressed in lines 301-310:
This is further supported by the fact that the most common reason for filing a lawsuit against a physician is not medical error, but rather a breakdown in the patient-physician relationship, primarily characterized by unsatisfactory communication between patients and physicians [33]. This is additionally confirmed by another study - poor communication with the physician was cited as a key reason for filing a lawsuit, with common complaints including a lack of information about potential health issues (70%), feeling misled (48%), unanswered questions (32%), and ignored concerns (13%). Similarly, a review of plaintiff depositions showed that 71% of cases involved communication failures, such as not listening to patients or providing clear explanations about adverse events. The authors even suggest that communication failures were more prevalent than originally recognized. A survey by Charles Vincent and colleagues of over 200 patients engaged in malpractice litigation also highlighted dissatisfaction with post-incident communication [34].
Comments 7: 3.4 probably, as 3.2, it could be useful to add tables to increase readability.
Response 7: We have included three additional tables two of the detailing the results of the chi-square test (lines 245 and 248), as well as familiarity and participation in mediation (line 205).
Response 8: We have further discussed the role of the judicial system in healthcare conflict resolution, added more references, and outlined the advantages of mediation (lines 318-338).
The plaintiff must prove the defendant's fault, and the judge rules based on evidence, witness testimonies, expert opinions, and the law. Strict procedures and rules make it difficult for plaintiffs, especially patients [36]. Furthermore, expert witnesses are crucial in healthcare cases, but many physicians refuse to serve as one due to heavy workloads, delays, time constraints, and unfamiliarity with court procedures [37]. As a result, many victims of medical errors do not receive compensation, and even when awarded, the amount is often unpredictable. Additionally, patients may have to pay attorneys up to 30–40% of the compensation. Healthcare litigation is also costly: legal expenses in the U.S. exceed $55 billion annually, while legal disputes in healthcare in the United Kingdom account for 0.04% of its gross domestic product [9,35,38,39]. Furthermore, high administrative and legal costs are a barrier for many patients to file a lawsuit. In Canada, the cost of initiating a lawsuit can reach between $40,000 and $50,000 [40]. Mediation, on the other hand, is cheaper, requires less preparation, and often eliminates the need for lawyers [41]. Another significant disadvantage of the traditional judicial system is the lengthy time required to make a ruling and conclude a case. According to Jury Verdict Research, the average time to resolve a medical malpractice case in the United States in 2000 was 45 months. In Malaysia, the duration of a medical malpractice trial can take a minimum of 15 years and may extend up to 25 years. In Canada, the time from filing a claim to reaching court is typically 5 to 6 years. In contrast, a study that included 13 mediation firms in the U.S. found that the average time to reach a resolution through mediation is just 1 to 3 days [11, 40, 42].
Comments 9: conclusions: to be weighed as the article seems more focused on information and communication issues while are limited those on mediation (60% are unaware of mediation)
Response 9: The conclusion has been revised as per your recommendation. It has been also shortened following the suggestion of another reviewer (lines 239-400):
This study highlights the potential of mediation as a valuable alternative to litigation in resolving healthcare disputes. While communication barriers between physicians and patients are a significant concern, with many patients unaware of mediation options (60% in this study), the results underscore the importance of raising awareness about mediation as a means to improve healthcare outcomes. The lack of trust and limited knowledge of mediation are significant obstacles that hinder its wider adoption. Future efforts should prioritize educating both healthcare professionals and patients about the benefits of mediation, integrating it into healthcare systems to foster better conflict resolution and strengthen trust in the healthcare system.
Comments 10: institutional review board: please remove sentence on anonymization, as difficult to sustain and irrelevant.
Response 10: The term "anonymized" was removed, and the sentence was rephrased as follows: "No personal data was retained and analyzed."
Comments 11: informed consent: rephrase as informed consent is linked with participation in the research, even sociological, and then for data management.
Response 11: Thank you for your valuable comments. The informed consent section has been revised as follows:
Informed Consent Statement: Participants provided informed consent.
Round 2
Reviewer 1 Report
Comments and Suggestions for Authors
We thank the authors for addressing all feedback. No additional changes are needed.
Author Response
Dear Reviewer,
Thank you for your thoughtful comments and feedback. We greatly appreciate your insights, which have contributed to improving the quality of our manuscript.
Reviewer 2 Report
Comments and Suggestions for Authors
Thank you to the authors for their efforts in addressing the comments raised in the first round of revision.
The manuscript is much better shaped than the original submission. However, a critical problem and two minor issues remain and need addressing.
The critical problem:
1- In its current presentation, the manuscript describes an unethical study and should not be accepted for publication. International standards/guidelines on human research, including online surveys, require the study to be ethically approved by a Research Ethics Committee or Institutional Board Review BEFORE starting the project. Unless the authors acquired ethical approval prior to this study or can guide the international readership that studies based on online surveys are exempt from obtaining IRB ethical approval, I would strongly recommend against the publication of this study.
Minor issues:
2- Table 3 is entirely illegible.
3- The response to comment number 7 must be included in the manuscript.
Author Response
Dear Reviewer, thank you for your valuable feedback. Your insightful comments have significantly strengthened the manuscript.
Comments 1- In its current presentation, the manuscript describes an unethical study and should not be accepted for publication. International standards/guidelines on human research, including online surveys, require the study to be ethically approved by a Research Ethics Committee or Institutional Board Review BEFORE starting the project. Unless the authors acquired ethical approval prior to this study or can guide the international readership that studies based on online surveys are exempt from obtaining IRB ethical approval, I would strongly recommend against the publication of this study.
Response 1 - Thank you for your message and for updating the submitting author’s name in the system. I sincerely apologize for the delayed response.
Our study was sociological in nature, aiming to examine participants' perceptions and the feasibility of mediation in conflict resolution in healthcare. The study did not pose any physical risk or psychological burden to participants. As it was conducted entirely online, we had no direct personal contact with participants, and all responses were collected anonymously, without gathering any personal information or identifiable details. Additionally, participants were fully informed about the study’s purpose before beginning the survey and provided their voluntary consent. They also had the option to withdraw at any time.
- According to the Bulgarian Health Act, a medical scientific study is defined as “each experiment on people, which is implemented with the objective of increasing medical knowledge” (Art. 197, para. 2). Ethics committee approval is required only for studies that meet this definition and involve experiments on human subjects (Art. 203, para. 1). Since our study did not involve any experiments, clinical interventions, or direct interactions with participants, and only collected anonymous data without any identification details, it does not fall under this category. Furthermore, as our study was sociological in nature, focusing solely on participants’ perceptions of mediation in healthcare, we do not believe it meets the criteria requiring ethics approval. The full legal reference can be found here: BGR-CC-31-01-LAW-2018-eng-Health-Act.pdf
- Our study consisted solely of an anonymous online survey, without the collection of personal or sensitive data. Given the absence of identifiable information and the lack of any potential harm to participants, the study posed minimal risk and aligns with the exemption criteria you mentioned for research using publicly available or non-identifiable data.
Based on these points, we believe that ethics committee review was not necessary for our study.
Comments 2 - Table 3 is entirely illegible.
Response 2 - We have revised the formatting and presentation of Table 3 to enhance its readability.
Comments 3- The response to comment number 7 must be included in the manuscript.
Response 3 – The response to comment 7 has been incorporated into the manuscript and can be found in lines 101-107.